# Discovering the Ultimate Limits of Protein Secondary Structure Prediction

**DOI:** 10.3390/biom11111627

**Published:** 2021-11-03

**Authors:** Chia-Tzu Ho, Yu-Wei Huang, Teng-Ruei Chen, Chia-Hua Lo, Wei-Cheng Lo

**Affiliations:** 1Institute of Bioinformatics and Systems Biology, National Yang Ming Chiao Tung University, Hsinchu 300, Taiwan; chiatzu.bt06@nycu.edu.tw (C.-T.H.); freddy789.bi07g@nctu.edu.tw (Y.-W.H.); ymchen.bi04g@g2.nctu.edu.tw (T.-R.C.); s109080506@m109.nthu.edu.tw (C.-H.L.); 2Department of Biological Science and Technology, National Yang Ming Chiao Tung University, Hsinchu 300, Taiwan; 3The Center for Bioinformatics Research, National Yang Ming Chiao Tung University, Hsinchu 300, Taiwan

**Keywords:** protein secondary structure prediction, protein sequence, protein structure, protein sequence-based predictions, structural biology

## Abstract

Secondary structure prediction (SSP) of proteins is an important structural biology technique with many applications. There have been ~300 algorithms published in the past seven decades with fierce competition in accuracy. In the first 60 years, the accuracy of three-state SSP rose from ~56% to 81%; after that, it has long stayed at 81–86%. In the 1990s, the theoretical limit of three-state SSP accuracy had been estimated to be 88%. Thus, SSP is now generally considered not challenging or too challenging to improve. However, we found that the limit of three-state SSP might be underestimated. Besides, there is still much room for improving segment-based and eight-state SSPs, but the limits of these emerging topics have not been determined. This work performs large-scale sequence and structural analyses to estimate SSP accuracy limits and assess state-of-the-art SSP methods. The limit of three-state SSP is re-estimated to be ~92%, 4–5% higher than previously expected, indicating that SSP is still challenging. The estimated limit of eight-state SSP is 84–87%. Several proposals for improving future SSP algorithms are made based on our results. We hope that these findings will help move forward the development of SSP and all its applications.

## 1. Introduction

Protein secondary structure prediction (SSP) means to predict the per-residue backbone conformation of a protein based on the amino acid sequence. It is an essential structural biology technique with a variety of applications. New SSP algorithms have been published almost every year for seven decades, and the competition for accuracy has always been fierce [1]. Recently, it is generally believed that the accuracy of current SSP methods has approached the theoretical upper limit of 88%, and it may be difficult to make breakthroughs; thus, the intensity of research and development in SSP seems weakened (see Figure 1 of [1]). Nevertheless, we found that the limit of SSP estimated by previous studies might not be precise enough and seemed underestimated. We believe that precisely determining the limit of SSP will help re-energize this field, set new directions for SSP developments, and ultimately benefit all applications relying on SSP. In this study, we performed exhaustive pairwise sequence and structure alignments on protein structure databases and estimated the theoretical limits of three- and eight-state SSPs. Besides, the experimental results revealed valuable information for future SSP developments.

Theoretically, it is possible to predict the three-dimensional structure and function of a protein through its sequence. However, such prediction is still challenging and time-consuming. A simplified strategy predicts the secondary structure based on sequence and uses the sequence and predicted secondary structural information to predict its 3D structure. Recently, genomes have been rapidly sequenced, and a large number of protein sequences are determined so that the number of proteins with known sequences exceeds the number of proteins with known structures by more than a thousand times. SSP is particularly important in this postgenomics era because we may use it to predict the structure and function of proteins rapidly and achieve many applications. For example, identification of disease-causing genetic mutations [2,3,4], homology detection and multiple alignments of proteins [5,6,7], evolutionary analyses [8,9,10], folding studies [11,12], and the prediction of enzyme target sites [13,14], functional sites [15,16], binding sites [17,18,19], and bioengineering sites [20,21,22,23]. Determinations of epitopes [24,25,26], protein-protein interaction surfaces [27,28], disordered regions [29,30,31,32], and protein subcellular localization [33,34] utilize SSP as well. Predicted secondary structures can be used in de novo protein design [35,36] and drug design [37,38,39]. Perhaps these wide applications are the main reason why, even when SSP accuracy has been considered to approach the limit, there have still been around five methods proposed every year since 2010 [1].

Before carrying out SSP, protein local backbone conformations were first classified into secondary structure elements (SSEs). There are two general sets of SSE: the three-state SSEs that describe protein conformations as helixes, strands, and coils, and the eight-state SSEs defined by the DSSP (Define Secondary Structure of Proteins) algorithm [40]. Hence, the development and assessment of SSP methods also fall into two categories, i.e., three-state predictions with Q3 accuracy or eight-state predictions with Q8 accuracy.

The SSP research field has been initiated since Pauling and Corey proposed the helix and sheet conformations of polypeptide chains roughly 70 years ago (1951) [41,42]. In the 1970s, amino acid propensities and residue or segmental physiochemical properties of polypeptides were used to perform SSP based on statistical approaches and achieved ~60% Q3 accuracy [43,44]. Since the late 1980s, as machine learning techniques were applied, the scale of the SSP feature set had been greatly expanded by adopting more residue properties and wider “window” (segment) sizes, and the Q3 was pushed to ~70% [45]. In 1999, a neural network machine learning SSP method, Psipred, first utilized the position-specific scoring matrix (PSSM) generated by PSI-BLAST [46] to be the feature set and accomplished a Q3 of 76.5% [47]. Since the big jump made by Psipred, the basic procedure of modern SSP methods has been established, which utilizes the PSSM as the main feature set and machine learning as the core algorithm. The main competition for accuracy then shifted to improvements in machine learning. The Q3 of methods developed in the early 2000s was 76–78% in general [48]. Meanwhile, the Q8 of SSpro8 was 62.6% [49]. When Dor and Zhou developed the SPINE using neural networks with a large amount of training data, the Q3 reached 79.5% in 2007 [50]. Soon after that, another neural network-based method, Jpred 3 [51], broke the 80% record of Q3 in 2008. Neural network-derived algorithms hence became the core of most SSP methods. In recent years, the architecture of neural networks used in SSP has become increasingly complex, including the BRNN [52], LSTM-BRNN [53,54], CNF [55], CRNN [56], and many deep learning techniques [57,58,59]. However, the progress of SSP accuracy seems to have encountered a plateau. Compared to the earliest sixty years, the improvement in accuracy during the past decade has been limited. The Q3 of current SSP methods, tested with independent datasets (<25% sequence identity with the training data), is between 81% and 86%, while the accuracy of Q8 is not yet above 77% [54,60].

Interestingly, even when SSP accuracy was still rapidly increasing, there had been studies estimating the upper limit of Q3 to be ~88% [61,62]. The ultimate reason for the existence of an upper limit in SSP is that proteins are not static/rigid bodies but dynamic objects with some parts being more flexible than others [48]. A protein may adopt different conformations under different conditions; besides, conformational changes commonly occur when interacting with ligands, substrates, other proteins, or cellular components. Thus, the conformation of a protein determined with different pH values, temperatures, metal ions, or cofactors would not be identical. Moreover, structures of the same protein solved by different techniques may have noticeable differences due to experimental equipment or conditions. For instance, X-ray diffraction requires crystallization of the protein, a relatively rigid and stable state, while nuclear magnetic resonance (NMR) spectroscopy determines protein structure in the solution phase, a state so flexible that an NMR experiment typically produces an ensemble of models. Inevitably, resolution, errors, and software inconsistencies are responsible for the limit of SSP, too [1,48,62]. Another type of limitation lies in how SSEs are assigned. Automated secondary structure assignment programs define SSEs by hydrogen-bond pattern, geometric features, or expert knowledge, and inconsistencies between programs are often observed [63]. Some technical characteristics in SSP methods may also cause limitations. The general methodology applied by state-of-the-art SSP methods works based on PSSM, implying their core assumption is that homologous proteins have similar structures. Indeed, it has been known that proteins sharing similar sequences have similar 3D structures [64]. However, being similar does not mean being identical. Therefore, in the current homology-based SSP methodology, the fundamental reason for the limit of accuracy is the structural difference or inconsistency between homologs.

Previous studies on the limit of SSP made estimations by measuring the secondary structural (in)consistency between structural homologs. In 1993, Levin et al. analyzed seven structural families (82 proteins sharing >17% sequence identities) from a protein structure database [65] and concluded that those structural homologs had only 88.8% secondary structural consistency [61]. In 1994, Rost et al. analyzed 140 pairs of structures sharing >30% sequence identities and reported that homologous structures differed by 12% in their three-state secondary structures, and thus the upper limit of Q3 should be 88% [62]. They also proposed a new accuracy measure termed SOV (segment overlap), which calculated the secondary structural consistency based on segments instead of residues. The upper limit of the first proposed SOV (v’94) was estimated to be 90 [62], which was then revised downward when an updated SOV (v’99) was proposed (see Table I in [66]). Yang et al. used PSI-BLAST to retrieve homologs from the Protein Data Bank (PDB) for a set of 1199 representative structures and classified those homologs according to sequence identities. They concluded that the limit of Q3 is >88% for structural homologs sharing ≥30% sequence identities [1].

After reviewing these pioneering works, we noticed that the limit of SSP can be more precisely estimated and that several estimates for evaluating current SSP methods are still in need. First of all, due to the amount of data available at that time, the earliest studies only analyzed a small number of proteins. Second, the secondary structural differences or consistencies of structural homologs were measured by sequence alignments, or by structural alignments with algorithms that are relatively primitive compared to today. Large-scale structural alignments were time-consuming, yet if sufficient computing power were available, it would be more appropriate to measure structural homologs’ structural differences/consistencies by structural alignments. Moreover, previous studies had only analyzed Q3 and the first SOV3 (v’94). The limits of Q8 and the new SOV3/8 (v’99) have not been estimated. Despite their high performance in residue-based three-state predictions, current SSP methods are still weak in segment-based and eight-state predictions [55]. The competition of SSP accuracy is about to switch to Q8 and SOV. These estimates shall be very informative for future SSP developments.

To precisely estimate the limit of SSP, large-scale analyses with multiple repeats were conducted in this study. Over ten times the number of PDB structures (~15,000) were used to repeat the experiments of previous studies. Also, exhaustive pairwise alignments were applied to 3192 structural families (~170,000 proteins) of the SCOP (Structural Classification of Proteins) database [67]. Four sequence and three structural alignment algorithms were recruited to measure the secondary structural (in)consistency between homologs. Figure A1 in Appendix B provides a schema of this study and the presented figures. Our results revealed that the theoretical upper limits of Q3 and v’99 SOV3 might be both revised upwards to 90–92%. As for eight-state measures, the upper limit of Q8 is 84–86%, and v’99 SOV8 is 85–87%. However, these **theoretical limits of SSP** should be challenging to achieve. Considering the current homology-based methodology, we proposed that the **practical limits of SSP methods** that work with PSI-BLAST PSSMs and PSSM reference datasets of <90% sequence identities would be Q3: 87.0 ± 1.2%, SOV3: 86.6 ± 1.5%, Q8: 78.8 ± 1.6%, and SOV8: 79.6 ± 1.6%. The results are divided into theoretical and practical limits because, although SSP limits can be accurately estimated by analyzing structural differences between structural homologs using structural alignments, in practice, current SSP methods can only rely on the homology detected by sequence alignments (structure of the query protein is unknown in an actual SSP). After further investigation, we found that the restriction caused by this “sequence-only” scenario may be conquerable, and specific suggestions have been made in the Discussion.

This study aimed to precisely estimate SSP limits and provide useful information for improving future SSP methods. Hopefully, our discoveries will help push SSP out of the accuracy plateau and eventually move forward all research and application fields depending on accurate protein secondary structure predictions.

## 2. Materials and Methods

### 2.1. Experimental Datasets

#### 2.1.1. Datasets Prepared from the Protein Data Bank

In the review by Yang et al. [1], a nonredundant (nr) query dataset of 1199 proteins and a PSSM reference dataset of all known protein structures were obtained from PDB 2005. In our work, the applied PDB data were released in 2015 Dec. The nr query dataset was prepared by downloading the 100% sequence identity representative PDB entity list of 2015 established by NCBI [68,69] (73,073 polypeptides; termed nrPDB100-2015 in this report) and then reducing the sequence identity from 90%, 80% … to 30% by three algorithms to ensure low redundancy of the final query proteins: CD-HIT [70], USEARCH [71], and MMseqs2 [72]. The produced nr datasets were named nrPDB90-2015, nrPDB80-2015, and so on. In the first experiment of Results, the 30% identity nonredundant dataset nrPDB30-2015 (15,059 protein entities; see Appendix A) was used as the query dataset, and the whole 2015 Dec. PDB (PDB-2015; 273,920 entities; Appendix A) was applied as the PSSM reference set. Other nr datasets were utilized to test how the homology of developmental datasets affects the accuracy of SSP methods.

#### 2.1.2. Homolog Data Obtained from the SCOP Database

Homologous protein structural data of SCOP were obtained from the SCOPe (v.2.07), which hierarchically classified protein structural domains into classes, folds, superfamilies, and families [67,73]. The four major classes were used, a: all-alpha, b: all-beta, c: alpha/beta (alpha-beta mixed), and d: alpha+beta (alpha-beta segregated). After (1) removing proteins with homology classifications only by automated matching procedures but not yet by SCOPe manual verifications, and (2) discarding entities unable to be correctly processed by DSSP, 169,977 domain entities were retrieved from 3192 families, which belonged to 927 folds. We termed this dataset SCOP-2.07 (see Appendix A). To examine the performance of current SSP methods on proteins of different structural classes or sizes, we prepared a query dataset from the SCOP-2.07 following these steps: (1) reduce the sequence identity of SCOP-2.07 to 30% using the homology reduction methods mentioned above. (2) Classify the 30% identity representative sequences into four subsets according to SCOP classes. (3) From each class subset, randomly select 60 proteins whose sizes are small (<150 residues), medium (150–299 residues), or large (≥300 residues). (4) Collect the selected proteins into a dataset, which hence comprise the same amount of proteins from the four major SCOP classes and the three size groups. The dataset prepared by this procedure contained 720 proteins (i.e., 4 × 60 × 3) and was called the SCOP720 dataset.

#### 2.1.3. The Source of PSSM Reference Sequences

For verifying whether our results properly estimated SSP limits, several state-of-the-art SSP methods were used to perform the tests described in Discussion. In those tests, the source of PSSM reference sequences was the 90% identity nr UniRef90 dataset established by Universal Protein Resource [74] in 2015 (UniRef90-2015; 38.2 million proteins). When necessary to ensure that the homology between the reference and query datasets was lower than some sequence identity, the interdataset homology reduction software CD-HIT-2D [70] was applied. Note that a PSSM reference dataset is typically termed a “target dataset/database” in fields like sequence similarity searches. However, in many protein structural prediction fields, a “target sequence” means the query sequence for prediction. To avoid confusion, we termed a target dataset a reference dataset in this report.

### 2.2. Assignment of Protein Secondary Structure

The secondary structure of PDB and SCOP structure entities was computed by DSSP [40] (release date: 1 April 2000), which assigned for each residue an SSE code. The DSSP classified protein backbone conformations into α-helix (H), 3_10_-helix (G), π-helix (I), extended strand participating in β ladder (E), isolated β-bridge (B), hydrogen-bonded bend (S), turn (T), and loop/irregular (C). In most SSP works, this eight-state classification was reduced into a three-state version, where the codes H, G, and I were represented by H (helices), E and B by E (extended strands), and the others by C (coils/loops). The three-state SSE codes used in this study were converted from the DSSP eight-state codes following those rules. It is noteworthy that the program of DSSP was modified in 2015 (v2.2.1) such that much more π-helices would be assigned [75]. This study and all the applied SSP methods were established based on the DSSP released before 2015. Because the occurrence of π-helices is very low in protein structures, the influence of the DSSP version on the estimated SSP limits and assessed accuracies of SSP methods is negligible. However, for some specific research topics where the correct assignment of π-helices is critical, attention must be paid to the choice of DSSP version.

### 2.3. Experimental Procedures

#### 2.3.1. Estimating SSP Limits by Structural Homologs Identified from the PDB Datasets

Following the procedure of [1], homologous protein structures were identified using the PSI-BLAST sequence alignment algorithm to search a PDB reference dataset against a set of query sequences for homolog pairs whose E-value was lower than a cutoff of 10^–3^. The reference dataset used in this study was PDB-2015, and the query set was nrPDB30-2015. Since nrPDB30-2015 was more than 10 times larger than the 1199-protein query set applied in [1], we were able to perform multiple repeats in this experiment. The nrPDB30-2015 was randomly divided into 10 subsets, each with 1500 proteins (remnants discarded). These query sets were used to perform PSI-BLAST searches from PDB-2015 for query-reference homolog pairs, which were then classified according to the sequence identity computed by PSI-BLAST into a series of identity levels, 0, 5, 10, …, and 100%. For each level, in addition to PSI-BLAST, homolog pairs were aligned by the structure alignment algorithm FAST [76]. Based on these sequence and structure alignments, the percentages of the secondary structural consistency between all pairs were averaged. Standard deviations of the average percent consistencies of the 10 repeats were also computed. The secondary structural consistency between homologs was calculated in two ways: the three-state and eight-state consistencies, equivalent to the Q3 and Q8 accuracies widely used in SSP research (see Equation (1) in Section 2.8.1).

#### 2.3.2. Estimating SSP Limits by Structural Homologs Determined by the SCOP Database

Proteins classified in the same SCOP fold possess similar composition of secondary structures, structural architecture, and topology; proteins in the same family are closely related homologs with the same functions and clear evidence for their common evolutionary origin [67,77]. There were 927 folds comprising 3192 families in the SCOP-2.07 dataset, and we randomly divided them into 10 subsets for multiple repeats of the experiment. We performed all-against-all pairwise sequence and structure alignments for each family by several algorithms. According to the alignments, the percent secondary structural consistency between each pair was computed. The secondary structural consistencies of homolog pairs belonging to the same fold were weighted averaged according to protein sizes to represent the fold. The secondary structural consistencies of all folds belonging to the same SCOP-2.07 subset were finally averaged to represent the subset. After applying the same procedure to all subsets, the standard deviation of these 10 repeats was calculated. The largest families in the SCOP-2.07 were composed of 3000 to 6000 protein entities. All-against-all alignments were very time-consuming. For instance, ~38.3 million alignments were made for the C1 set domain family (SCOPe ID: b.1.1.1; 6190 entities). For efficiency, the distributed computation system we developed for the *i*SARST protein structural similarity search system was utilized [78].

### 2.4. Applied Sequence Alignment Methods

The sequence alignment methods utilized in this study included local alignment algorithms PSI-BLAST (v.2.3.0) [46], the classic BLAST (v.2.2.13) [79], and Water [80], as well as the global sequence alignment algorithm Stretcher [81]. In the experiment on PDB, the PSI-BLAST was applied using its default parameters except for the E-value cutoff (set as 10^−3^, stricter than the default value 10). To ensure all applied algorithms were executed on an equivalent basis, their gap opening and extension penalties were uniformly set to be 11 and 2, respectively. Water was an implementation of the Smith-Waterman algorithm [82], and Stretcher was developed based on the Needleman–Wunsch algorithm [83]; both programs were obtained from the EMBOSS server (v.EMBOSS:6.6.0.0) [80,81].

### 2.5. Applied Structure Alignment Methods

Structure alignment algorithms including FAST [76], TM-align [84], and SARST [85] were applied. The programs of these algorithms accepted single-chain and single-model protein structure entity files in the PDB format. To apply them to the structure entry files obtained from the PDB database, when an entry file contained multiple chains, the chain of interest was extracted and saved in an individual chain entity file. When a chain entity had multiple models, only the first was used. As for domain entities from the SCOP database, the preprocessing was more complicated because a SCOP domain might comprise polypeptide fragments from multiple chains. Before a SCOP domain entity file containing multiple fragments could be accepted by these alignment algorithms, (1) the chain identifiers of the fragments were unified as “A,” (2) all residues of the entity were renumbered according to the sequence they appeared in the structure file, and (3) all atoms of the entity were also renumbered.

### 2.6. Applied Secondary Structure Prediction Methods

Several state-of-the-art SSP methods were applied to help verify the estimated upper limits of SSP, inclusive of three-state algorithms Psipred v3.3 [47], SpineX v2.0 [86], Scorpion v1.0 [87] and Spider2 v2.0 [88], and eight-state algorithms RaptorX v1.0 [89], SSpro8 v5.2 [90] and DeepCNF v1.02 [55]. The utilized versions of these methods were all trained and released before 2016. Their original pipelines relied on different editions and settings of PSI-BLAST to generate PSSM. To make all methods applied on an equal basis, we modified their pipeline scripts to unify the edition and settings of the PSI-BLAST engine (v2.3.0) [46]. The unified PSI-BLAST settings were based on the common parameter settings used by most of these SSP methods and the defaults recommended by PSI-BLAST (see Appendix A for the original and modified settings). To ensure that we correctly applied these methods and the modified settings of PSI-BLAST would not hinder their performance, we preliminarily conducted SSP using these state-of-the-art methods with UniRef90-2015 as the PSSM reference dataset. The query sets were the TS115 and CASP12 prepared by Yang et al. [1], which were consisted of novel PDB structures released later than 1 January 2016, meaning that the machine learning core of the applied SSP methods had not learned those proteins. In particular, the CASP12 was obtained from the competition of the 12th biannual meeting of Critical Assessment of Structure Prediction techniques [91]. Results listed in Table 1 showed that the accuracy values computed using our pipelines were all close to the values obtained from previous reports of these SSP methods assessed with equivalent datasets [1,55].

### 2.7. Implemented Secondary Structure Prediction Models

In implementing the general SSP methodology (see Section 4.3 of Discussion), the PSI-BLAST-generated PSSM utilized by current SSP methods and the amino acid information used in many recent SSP works [55,92,93,94,95] were applied as the predictive feature set. For each residue, the scale of the feature set was expanded by referring to the features of its nearby residues based on the “window” concept proposed in the Psipred algorithm [47]. The window size was set at 5. The SSP models were trained with the integrated machine learning system we developed [21,22]. The integrated algorithms included artificial neural networks, decision trees, support vector machines, and bootstrap sampling. Settings of these algorithms were according to the defaults of this system [22].

### 2.8. Computation of Accuracy Measures

#### 2.8.1. The Q Accuracy Measure and the Percent Secondary Structural Consistency between Homologous Protein Pairs

For measuring the secondary structural consistency between a pair of homologous proteins, the secondary structures of the two proteins were first represented by their per residue SSE sequences generated by DSSP. Next, according to the alignment between the proteins, the percent consistency (C_3_ or C_8_) in SSE sequences was calculated for all aligned equivalent residues using the following equation:(1)$$C_{s}=\frac{n_{c}}{n_{e}}\times 100\;\left(\%\right),\;\;s=\left\{\begin{array}{c} 3,\;\mathrm{for}\;\mathrm{three}-\mathrm{class}\;\mathrm{SSE}\;\mathrm{codes}\\ 8,\;\mathrm{for}\;\mathrm{eight}-\mathrm{class}\;\mathrm{SSE}\;\mathrm{codes} \end{array}\right.$$
where *n_e_* stands for the number of equivalent residue pairs based on the alignment, and *n_c_* represents the number of equivalent residue pairs having the same SSE codes. This definition of percent SSE consistency is equivalent to the SSP accuracy measure Q, defined as the number of correctly predicted residues divided by the total number of predicted residues. In this report, the C_3_/C_8_ measures are simply written as Q3/Q8.

#### 2.8.2. The SOV Measure

In addition to the Q measure, we also estimated the limit of SSP accuracy using the SOV v’99 edition [66]. SOV was developed for evaluating the consistency between the predicted and actual SSE sequences of a protein based on secondary structure segments instead of residues. It has been generally regarded as a more critical accuracy measure than Q for its capability of capturing the overall consistency of two SSE sequences and reducing noises from individual residues [62,66]. Most studies calculated SOV based on the three-state SSEs, i.e., the SOV3. In this work, the eight-state SOV (SOV8) was also calculated.

#### 2.8.3. The Weighted Average of Accuracy Measures

Protein pairs may have quite different sizes. If the classic arithmetic mean were applied to compute the average Q or SOV (i.e., *sum/n*, where *sum* is the summation of measure values from all pairs and *n* denotes the number of pairs), the influence of small proteins might be overestimated while that of large proteins be underestimated. In this study, the average Q and SOV were calculated as the weighted average, with the number of aligned residues for each protein pair used as the weight, that is:(2)$$\bar{Q}=\frac{\sum_{i=1}^{N}n_{e,i}\times Q_{i}}{\sum_{i=1}^{N}n_{e,i}}$$
(3)$$\bar{\mathrm{SOV}}=\frac{\sum_{i=1}^{N}n_{e,i}\times SOV_{i}}{\sum_{i=1}^{N}n_{e,i}}$$
where $\bar{Q}$ and $\bar{\mathrm{SOV}}$ represent the weighted average of Q and SOV, *N* denotes the number of protein pairs, *n_e_*_,*i*_ means the number of aligned equivalent residue pairs in the *i*th protein pair, and *Q_i_* and *SOV_i_* represent the measure values of the *i*th protein pair.

## 3. Results

### 3.1. Experiments Based on PDB Datasets

The procedure Yang et al. estimated the upper limit of SSP [1] was repeated to verify whether we correctly applied all algorithms and check whether the conclusion of [1] was still applicable as the amount of protein structure data increased rapidly. A set of query structures was used to retrieve homologs from the entire PDB by PSI-BLAST sequence alignment searches [46]. The secondary structural consistency between homologs was computed for a given sequence identity level (individual) or identity threshold (cumulative). Since the query set we utilized was over 10 times larger than previous works’, 10 repeats of the experiment were thus allowed by random sampling.

As Figure 1 shows, both the curve for individual identity levels and the cumulative curve of our results were similar to those reported in [1], with minor differences at identities ~70%. The main reason for the differences was that Yang et al. discarded structures solved not by X-ray or with low resolution, but we did not. By applying the same restrictions, our curves got closer to theirs. However, to make conclusions applicable to a broader scope, we decided to keep using all available structures without restrictions on the method of structural determination or resolution. The data of this figure are listed in Table 2. When the identities between homologous structures were ≥30%, their secondary structural consistency was ≥87.8%, agreed well with [1,62]. The consistency was also analyzed by FAST structural alignment [76]. The results indicated that the limit of SSP could be higher than that estimated by sequence alignments, especially for homologs with low identities. Judging from the structure-alignment data, the limit of SSP was >90% as the identity of homologs was ≥30% (see identity level 30–35 for FAST in Figure 1 and Table 2).

Previous studies estimated SSP limits using three-state SSEs. We also used eight-state SSEs and estimated the Q8 accuracy, a more challenging assessment. In general, the Q8s achieved by state-of-the-art SSP methods are ≤77% [54,60]. Figure 1 and Table 3 demonstrate that when the sequence identities between homologs were >30%, the upper limit of Q8 was >79% as estimated by PSI-BLAST sequence alignments; when estimated by FAST structure alignments, the limit of Q8 was >83%. These data revealed that there is still much space left for improvement in eight-state SSP.

The three-state secondary structural consistency between a pair of proteins is analogous to the Q3 accuracy of SSP, if one protein were considered as the query and the other were the answer. The full three-/eight-state results of this experiment can be found in Appendix A.

To analyze the secondary structural consistency of homologs based on secondary structure segments, we computed the SOV for these PDB pairs. Computed by sequence alignments, the SOV3 was a little lower than Q3. Contrarily, the SOV3 computed by structural alignments was slightly higher than Q3 (see Appendix A). When it was first proposed, the upper limit of SOV3 (v’94) was estimated to be higher than that of Q3 [62]. However, the upper limit of the updated SOV3 (v’99) has not been determined until this work. Considering structural alignment is better than sequence alignment at detecting residue-residue correspondence between proteins [61,96], it was supposed that the FAST alignments were more reliable than PSI-BLAST alignments in estimating SSP limits, inclusive of SOV.

### 3.2. Limit of SSP Accuracy Estimated by Protein Sequence Alignments between SCOP-Determined Structural Homologs

In the previous experiment, the quality of homologs was controlled by the E-value cutoff of PSI-BLAST as it searched PDB ([1] and Materials and Methods). Now we modified the experimental design by using the structure families classified by SCOP [67] to define homologs. For each family, all-against-all pairwise sequence alignments were performed by PSI-BLAST [46], classic BLAST [79], Water, and Stretcher [80,81]. Based on these alignments, the secondary structural consistencies of homologs were computed. Figure 2a,b shows that no matter using homologs automatically identified by PSI-BLAST or semi-manually verified by structural biologists like SCOP, as long as the identities were ≥30%, the secondary structural consistencies between homologs were similar. As for identities <30%, the consistency of SCOP family homologs was obviously higher than that of PSI-BLAST identified homologs, especially when measured by PSI-BLAST, classic BLAST, or Water (see Appendix A for raw data).

Following previous works [1,62], we calculated the secondary structural consistency for homologs with identities higher than certain thresholds (e.g., Figure 1). However, preparing protein datasets with identities higher than a threshold is uncommon in structural biology applications. The typical way is making nonredundant datasets with identities lower than certain cutoff(s), just like the nrPDB30-2015, where any two proteins shared identities <30%. We supposed that the information about secondary structural consistency of homologs sharing identities below specific cutoffs would help estimate the practical limit of SSP methods. For example, results of such cutoffs shown in Figure 2c,d and Appendix A demonstrate that the highest average three-state secondary structural consistencies were 87.3% and 80.9% (registered by Water) for homologs with identities <90% and <30%, respectively. These consistencies suggested that when the sequence identity redundancy of the datasets utilized to develop SSP algorithms was <90% (or <30%), the operational upper limit of Q3 accuracy would be around 87% (or 81%). Speculated similarly, the upper limit of Q8 for SSP algorithms would be ~79% or ~70% if the sequence identity redundancy of developmental datasets were <90% or <30%, respectively (refer to the dotted lines in Figure 2c,d). See Discussion for more information.

### 3.3. Limit of SSP Accuracy Estimated by Protein Structure Alignments between SCOP-Determined Structural Homologs

Since the fundamental reason for the upper limit of SSP is the structural difference among structural homologs, we supposed that it should be more reasonable to estimate SSP limits using structural methods than using sequence-based methods. The above experiment on SCOP families was repeated here using structural alignment methods, FAST [76], TM-align [84], and SARST [85]. As Figure 3 shows, the secondary structural consistencies computed by structural alignments were higher than those by sequence alignments, especially when the identity was <30%. Speculating based on the results of these structural alignments, if the sequence identity redundancy of the developmental datasets for SSP was <90% or <30%, the maximally achievable average Q3 might be 90% or 86% (by FAST), and the maximally achievable average Q8 might be 83% or 78%, respectively. These upper limits estimated by structural alignments were 3–8% higher than those by sequence alignments, compared with Figure 2 (see also Appendix A for the results of SOV3 and SOV8).

### 3.4. Limit of Accuracy for Various Protein Structure Classes and Sizes

We divided the homologs from SCOP into four structural classes according to SCOP classifications [67] and observed the secondary structural consistency between homologs in each class. Besides, the homologs were divided into three size groups by sequence lengths. For simplicity, only the consistencies at 90% and 30% sequence identity cutoffs are displayed in Figure 4 (see Appendix A for complete results). No matter whether categorized by structural class or size, SOVs were generally higher than Q consistencies. Among the four classes, all-beta proteins had the lowest consistency. Among the size groups, large proteins (>300 residues) exhibited the highest consistency. In these plots, both the secondary structural consistencies computed by structure alignment (representative algorithm: FAST) and sequence alignment (representative algorithm: PSI-BLAST) were provided, demonstrating the gap between them, which we supposed was the difference between the theoretical limit of accuracy in SSP and the practical limit of accuracy for SSP algorithms.

### 3.5. The Lower Bound of Secondary Structure Prediction Accuracy

The experiments described above focused on the upper limit. In this part, we determined the lower bound of SSP. All domain entities of SCOP-2.07 were transformed into secondary structural strings composed of three- or eight-state SSE codes. A million protein pairs were randomly selected from this SSE string dataset, and the secondary structural consistencies (Qs and SOVs) between the random pairs were computed. Distributions of these measures are shown in Figure 5, and the detailed results and the background occurrence frequency of each three-/eight-state SSE are listed in Appendix A. The averages of Q3, Q8, SOV3, and SOV8 were 34.9%, 21.9%, 32.3%, and 22.0%, respectively.

## 4. Discussion

### 4.1. Difference between Using Homologs Identified by Programs and Those Determined by SCOP

Previous studies mainly analyzed the limit of SSP using homologous protein pairs identified by sequence alignment searches [1,62]. Rost et al. took some representative structures prepared by [97] as the core protein dataset and retrieved homologs sharing >30% identities to those proteins from PDB by sequence alignment [62]. Yang et al. used a set of 1199 representative structures as the core dataset and retrieved the homologs meeting a strict sequence alignment quality cutoff, i.e., E-value <10^–3^, by PSI-BLAST from PDB [1]. Both studies concluded that the limit of SSP would be >88% for homologs sharing >30% sequence identities. In the present study, homologs identified by PSI-BLAST from PDB based on a much larger amount of representative core proteins were applied and reached the same conclusion (Figure 1). However, there were some shortcomings in this experimental design. First, the logic behind this experiment was to measure the secondary structural differences of homologous “structures.” Determining whether proteins were homologous structures by sequence alignments seemed contradictory and might not be as precise as structural alignments. Second, the PSI-BLAST is a local sequence alignment method, meaning that even when two proteins were not related in evolution, it might still report some aligned fragments between them. PSI-BLAST provided a statistical significance index, the E-value, to help judge the quality of alignments; however, a strict E-value setting would only reduce the number and length of aligned fragments rather than ensure the structural equivalence between the fragments.

Using well-classified protein structural families can avoid these problems. This idea was also implied by Rost et al. [62] and had been implemented by Levin et al. in [61], where seven protein structural families prepared by [65] were used to quantify SSP accuracy. The present study utilized the structural families prepared by SCOP, which used automated screening and manual curation to classify proteins based on their structural, sequence, functional, and evolutionary similarities/relationships [67,77]. Entities in the same SCOP family are closely related global structural homologs with sufficient supporting data for their common origin [67,77]. Estimating SSP limits using SCOP family entities avoids the ambiguity in identifying structural homologs by sequence alignment search. The limitation of local alignments can be ignored because family entities are global homologs. As shown in Figure 2, for identities <30%, the SSP accuracies estimated using SCOP data were much higher than those estimated using PSI-BLAST identified homologs. We supposed that manually-verified structural homologs from a reliable source fit well the purpose of estimating the limit of SSP according to the structural differences of structural homologs and thus used the SCOP database to perform most experiments.

### 4.2. The Different Meanings between the Upper Limits of SSP Accuracy Estimated by Sequence and Structural Alignments

SSP is limited by the “structural differences” of homologous structures. Previous studies noted that sequence alignment had inherent inaccuracies compared to structure alignment when it was applied to calculating the secondary structural (in)consistency between homologous structures [62], but they still utilized sequence alignments. We speculated that this contradiction was because of some technical limitations. In the 1990s, the speed and precision of protein structural alignment algorithms were not as high as algorithms developed after the mid-2000s. Despite the high precision of algorithms nowadays, performing structural alignments is still much more time-consuming than sequence alignments. We did apply sequence alignments to compute the secondary structural consistency of homolog structures, and the results were very close to previous reports’ (Figure 1 and Figure 2; Table 2). Nevertheless, it seemed more appropriate that the structural differences of structures should be computed by structural alignments. For the preciseness of estimation, we utilized structural alignments. The heavy computing tasks were accomplished by an efficient distributed computation system [78]. Since structural alignment could determine the residue-residue equivalence in 3D space much better than sequence alignment, we expected that the SSP upper limit estimated by structure alignment would be higher than sequence alignment. Our results agreed well with the expectation (Table 2, Table 3, Appendix A). Take the 30–35% sequence identity level for example, the highest Q3 estimated by structural alignments was 89–90%, while estimated by sequence alignments was only 87–88% (Table 2 and Appendix A). We inferred that this difference was not an indicator that structural alignment is superior to sequence alignment in estimating SSP limits, but it implied something interesting:Accuracies estimated by structure alignment should be the ultimate **theoretical** upper limit of SSP because the way they got computed strictly followed the logic that structural homologs’ structural differences were measured by structural alignments.Accuracies estimated by sequence alignment is the **practical** upper limit of SSP since their computation folloTable Swed the procedure of the current SSP methodology.

In practice, only sequence alignment, but not structural alignment, is applicable in predicting the secondary structure for a query sequence. In other words, sequence alignments could not estimate the ultimate limit of “SSP” but could estimate the operational limit of “SSP methods.”

### 4.3. Information Revealed by the Estimated Limits of SSP under Different Sequence Identity Cutoffs

Previous studies on SSP limits made estimates for homolog structures with sequence identities **above given thresholds** (e.g., >30%). To our knowledge, the present study is the first to provide the estimates for homologs with identities **below given cutoffs** (e.g., <90% or <30%), supposedly helpful for knowing the limit of SSP under situations where only homologs sharing identities lower than some cutoffs can be used to make SSPs. Based on this supposition and Figure 2c, it was expected that the SSP limit estimated by PSI-BLAST at the 90% identity cutoff would define the upper bound of PSI-BLAST-driven SSP methods in which the sequence identities between (and within) the query and reference sets for training and testing were <90%. Similarly, if the sequence identities between/within the datasets were <30%, the estimates by PSI-BLAST at the 30% identity cutoff might define the upper bound of SSP accuracy.

We utilized three query datasets and seven state-of-the-art SSP algorithms to perform actual SSPs to verify the supposition. The datasets included the representative datasets prepared by Yang et al., the TS115 and CASP12, comprising structures deposited in PDB 2016 and highly nonredundant from those deposited before 2016 [1], and the SCOP720 prepared in this work. TS115 and CASP12 were particularly suitable because all the applied SSP algorithms were developed before 2016, meaning that their prediction models were not familiar with these 2016 novel proteins. The SCOP720 consisted of an equivalent number of small, medium, and large proteins from the four major structural classes (see Materials and Methods) and should achieve nonbiased evaluations. The source of PSSM reference proteins was the 90% identity nr dataset UniRef90-2015. Before the experiment, homology reductions were performed on UniRef90-2015 by CD-HIT-2D [70] to ensure that the reference dataset shared <90% identities with any protein in the query sets. The SSP methods were then applied using the same version of PSI-BLAST as the search engine. Figure 6 demonstrated that, in this 90% identity cutoff test, there was not yet any SSP method reaching the upper limits estimated by PSI-BLAST sequence alignments.

The homology between/within the datasets for “testing” was known in this test, but not the homology between/within those used to “train” the prediction models. Because of the unavailability of the source code of those SSP methods, we were unable to retrain their models. To further examine whether the secondary structural consistencies of homologs at various identity cutoffs can be used to estimate the limit of SSP methods trained/operated with protein datasets with different levels of homology reduction, we performed the following experiment. First, nr sets of sequence identity cutoffs 90%, 80%, …, and 30% were reduced from the PDB-2015 as described in Materials and Methods. Second, at each identity cutoff, from the corresponding nr dataset, 250 proteins were randomly selected as the training query set, another 250 as the testing query set, and 10,000 as the PSSM reference dataset. The reference dataset was small because the lowest identity dataset (i.e., nrPDB30-2015) contained only 15,059 proteins. Third, at each identity cutoff, a machine learning prediction model was trained with a PSSM-based feature set generated by performing PSI-BLAST searches against the reference dataset for all query sequences of the training set. Finally, by predicting the secondary structure of proteins in the testing set, accuracies were measured. As expected, results shown in Figure 7 revealed that as the homology of developmental datasets lowered, the accuracy of the established SSP model decreased, with a trend similar to the decreasing SSP accuracy limits estimated by PSI-BLAST. These results supported our idea that the accuracy estimated at different identity cutoffs may serve as a good indicator of SSP’s practical limit in research or applications where SSP is operated with homology-reduced datasets.

### 4.4. On the SSP Accuracies for Proteins of Different Structural Classes or Sizes and Residues with Different Physical Properties

To see how current SSP methods perform for different types of proteins and examine in detail our proposal that estimates made by structural and sequence alignments respectively stand for the theoretical and practical limits of SSP, we observed the accuracies of state-of-the-art SSP methods on different protein structural classes or sizes. The SCOP720 query set and UniRef90-2015 reference set were used. Figure 8 and Appendix A show that the accuracies of all tested methods were lower than the limits estimated by PSI-BLAST sequence alignment. All methods performed relatively low for all-beta proteins. Previous studies observed that among helices, strands (β-ladders and β-bridges), and coils, the accuracy of SSP methods is usually lowest in strands [1,88], perhaps due to difficulty in catching long-range interactions of residues forming β-sheet strands [1,48]. By estimating the upper limits of SSP for every eight-state SSE and examining for each SSE the prediction accuracy of state-of-the-art SSP methods, we clearly saw that, compared with helices, the performance of current methods is low for β-ladders and extremely low for β-bridges (see Appendix A for figures and Appendix A for detailed data). Since all-beta proteins mainly comprise β-strands, this class’s low accuracy seemed reasonable. Moreover, the Q3/8 of these methods were high for all-alpha proteins, but their SOVs were significantly lower than Qs. The difference between Q and SOV for all-alpha proteins was the largest among these classes. Considering SOV was designed to assess SSP by secondary structural segments [62,66], there seemed to be some weakness of current SSP methods in predicting the pattern of segments for proteins composed mainly of helices. The monotonic SSE composition of such proteins might be where the difficulty lies. For instance, between an actual SSE segment HHH**C**HHHCHHHC and a predicted HHHHHHHCHHHC, with the difference of just one position, the Q3 is 91.7% while the SOV3 is only 70.2%. Unlike structural classes, the SSP methods’ performances for different sizes were even, although the accuracy slightly decreased as the size increased. These results suggested that overcoming the difficulties brought about by long-range interacting β-sheets and monotonic helices codes shall be the key to improving future SSP algorithms.

In addition to protein-level classifications, we have also classified the aligned residues between SCOP family homologs according to some physical properties, for instance, solvent-accessible residues (≥25% RSA, relative solvent accessibility) vs. inaccessible ones (otherwise) based on the RSA calculated by the NACCESS [98], and residues having high vs. low B-factors by the normalization and threshold applied in [99]. As shown in Appendix A, we observed that solvent-inaccessible residues (buried region) have higher SSP limits than accessible ones, and residues with low B-factors (relatively inflexible) have higher SSP limits than those with high B-factors. Interestingly, judging by the estimated upper limits, the average accuracy of modern SSP algorithms is relatively lower for solvent inaccessible residues or residues with low B-factors than for solvent-accessible or flexible residues, revealing the direction toward improvements for future SSP algorithms.

### 4.5. Light Shed from the Discovery of This Study

Although structure alignments drew a higher upper bound on SSP accuracy than sequence alignment, the difference between the limits estimated by them was not significant until the identity of homologs became lower than 30–35% (Figure 3), implying that the alignment algorithm’s capability of determining the residue-residue equivalence between homologs with low sequence identity would be critical to improving SSP. This possibility might be supported by the fact that, for low-identity homologs, the secondary structural consistency measured by Water [80] was higher than that by PSI-BLAST or the classic BLAST (Figure 2 and Appendix A). Water is a full implementation of the Smith-Waterman dynamic programming [80,82], whereas (PSI-)BLAST are heuristic implementations much faster than the Smith-Waterman algorithm without guarantee of finding the optimal alignment [79]. To further test this possibility of SSP improvement, we repeated the sequence-alignment-based experiment of Figure 2 using (PSI-)BLAST, which provided a word size parameter to control the nucleation of alignments. A short word size could find more exact local matches between sequences, which might increase the amount of correctly detected residue-residue equivalences between structural homologs. The (PSI-)BLAST would most mimic the full Smith-Waterman dynamic programming with the word size 1 (default = 3). By decreasing the word size, we found the secondary structural consistencies reported by (PSI-)BLAST all raised (Figure 9), especially when the identity between homologs was low. These results supported that the accuracy of future SSP methods could be enhanced by improving the sequence alignment engine in determining the residue-residue equivalence between low-identity homologs. Drawbacks of this idea may include (1) a small word size increases the time cost, and (2) the shortest word size 1 is accepted only by the classic BLAST running in the pairwise mode but not PSI-BLAST nor BLAST in the batch searching mode, without which the alignment search against a large dataset like UniRef90 will be extremely time-consuming. Reprogramming accurate pairwise alignment methods like Water to support batch searching or modifying PSI-BLAST to accept the shortest word size may be part of the solution.

The difference between the secondary structural consistency measured by sequence and structure alignments implied that the nature of the methodology might restrict the operational limit of SSP. By “nature of the methodology,” we mean that the core procedure of current SSP methods is to generate the PSSM-based feature set by sequence alignment rather than structural alignment because the structure of the query is unknown. The ways out of this restriction may include, (1) updating the fundamental SSP feature set, for the PSSM-based set has lasted for 20 years without significant modifications (see [94] for a feature set proposed recently), (2) enhancing the balance of SSEs in the training data, for we observed that the SSEs with low occurrence were barely or never predicted by current SSP method, such as β-bridges and π-helices (Appendix A and Appendix A; see also [100] for similar observations), (3) redesigning the algorithm to be staged, for instance, by predicting the structural class of the query in the first stage and making the final prediction with a tailored model established for the class, (4) refining per-residue SSE predictions according to predicted physical properties of residues like solvent accessibility and backbone flexibility (see Appendix A), and (5) introducing sequence-structure relationship information generated by statistical techniques (e.g., the sequence structural entropy [101], and the per-residue flexibility encoded by local structural alphabets [100,102]) into the methodology to overcome the sequence-only scenario.

## 5. Conclusions

This study applied PDB and SCOP data, sequence and structural alignments, and random pairings to estimate the upper and lower limits of SSP accuracy, as summarized in Figure 10. Estimated by sequence alignments between homologous structures, the upper limit of Q3 is 87.0 ± 1.2 (%), agreed with previous studies. The upper limit observed in this situation is supposed to be the practical limit of the current SSP methodology, which works based on sequence homology encoded in the PSSM. Estimated by structural alignments between homologous structures, the upper limit of Q3 is 91.4 ± 0.8 (%). The upper limit obtained through this procedure is supposed the ultimate theoretical limit of SSP. Besides, the practical and theoretical limit of Q8 was estimated to be 78.8 ± 1.6 (%) and 85.0 ± 0.9 (%), respectively. The Q3 upper limit is ~4% higher than previously expected, indicating that developing SSP algorithms is still challenging. The Q8 accuracy of state-of-the-art SSP methods is ≤77%; hence, there is still 7–9% space to develop eight-state SSP algorithms. The gap between the practical and theoretical limits may be resolvable, and several proposals for breaking through the practical limits have been made in this report. We hope that these accuracy estimates can help set clear goals for future algorithm developments, and our discoveries will stimulate the SSP field and benefit all research and applications where protein secondary structure prediction plays a key role.

## Figures and Tables

**Figure 1 biomolecules-11-01627-f001:**
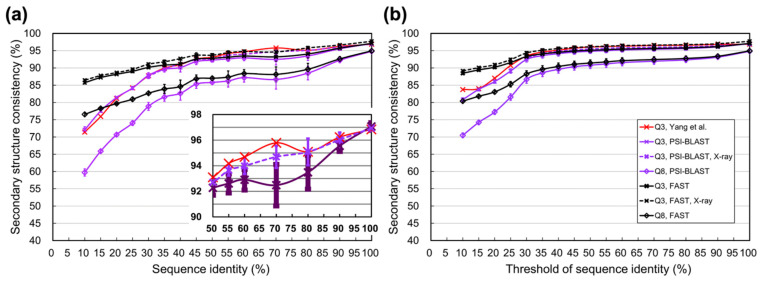
Secondary structural consistency between homologs identified by PSI-BLAST from PDB. (**a**) Secondary structural consistency at different sequence identity levels. The horizontal axis indicates the sequence identity level where, for instance, 30 means that the sequence identities of homologous structures fall between ≥30% and <35%. These levels were made according to Yang et al. [1]. The consistency dropped as the identity decreased, and our 10-repeat experimental results agreed well with [1] (the solid purple curve of Q3 versus the red). Yang et al. only analyzed high resolutions (<3 Å) X-ray structures. Following the same filtering criteria, our results (dotted purple curves) were closer to theirs. For easy observation, we enlarged the vertical axis of Q3 with identities ≥50. The secondary structural consistency revealed the structural difference between homologs, which is the ultimate factor limiting SSP accuracy. For instance, at the 30% identity level, the averaged consistency is 87.8%, meaning an 11.2% difference in secondary structure, which indicates that the SSP accuracy of such homologs can be at most ~88%. Previous studies measured the secondary structure consistency by sequence alignments [1,62]. We also performed structural alignments (black lines). (**b**) Secondary structural consistency at different sequence identity thresholds. The horizontal axis indicates the threshold where, for instance, 30 means that the sequence identities of homologous structures are ≥30%. Previous studies estimated the limit of SSP on the basis of three-state SSEs. We also made estimations based on the eight-state SSEs.

**Figure 2 biomolecules-11-01627-f002:**
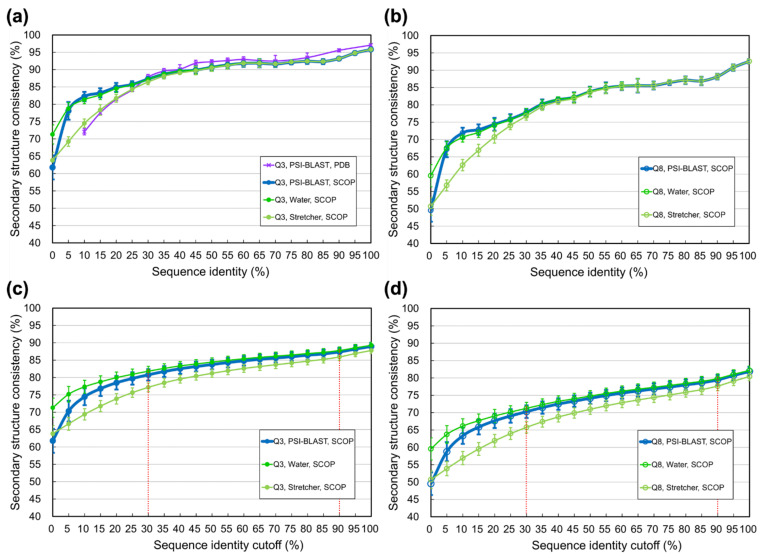
Secondary structural consistency measured by sequence alignments between structural homologs verified by SCOP. (**a**) Three-state secondary structural consistency (Q3) at various sequence identity levels. Using the structural homologs determined by SCOP to repeat the experiment of previous studies, the Q3 data of homologs sharing ≥30% identity were very similar to previous results. The purple curve of PSI-BLAST and PDB is obtained from Figure 1a for comparison. The Q3 differed significantly at low identities, with SCOP-determined homologs being much higher than PSI-BLAST-identified ones. Note that the difference was mainly caused by how homologs were defined (manual curations vs. programmatic searches) rather than by the source of data (SCOP vs. PDB). The Q3 was measured by PSI-BLAST, classic BLAST, Water, and Stretcher. The classic BLAST was omitted in this figure for clarity because its curve lay closely to that of PSI-BLAST (see Appendix A for full data, with the classic BLAST included). (**b**) Eight-state secondary structural consistency (Q8) at various sequence identity levels. (**c**) Q3 at various sequence identity cutoffs. The horizontal axis indicates the cutoff where, for instance, 30 and 90 mean that the identities of homologous structures are lower than 30% and 90%, respectively. Since protein datasets are typically prepared as nonredundant sets of identities lower than given cutoffs in practical research, in all figures after this, we only display the consistency computed under given identity cutoffs instead of thresholds. (**d**) Q8 at various sequence identity cutoffs.

**Figure 3 biomolecules-11-01627-f003:**
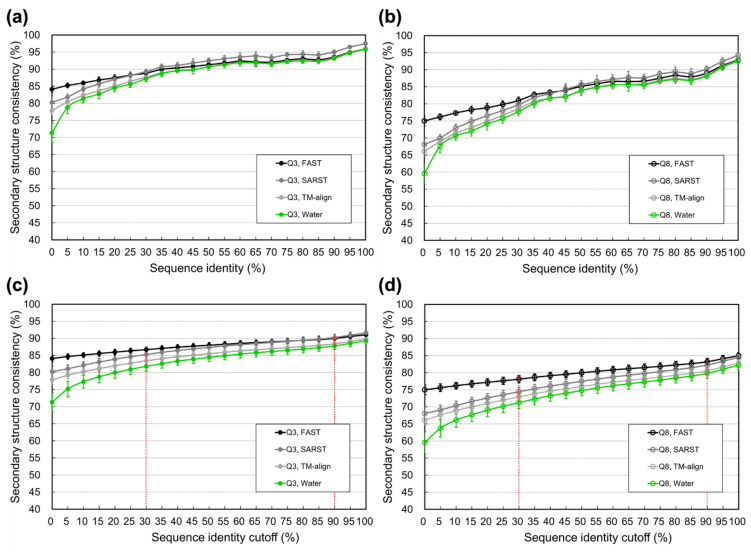
Secondary structural consistency measured by structural alignments between structural homologs determined by SCOP. (**a**) Three-state secondary structural consistency (Q3) at various sequence identity levels. (**b**) Eight-state secondary structural consistency (Q8) at various sequence identity levels. (**c**) Q3 at different various identity cutoffs. (**d**) Q8 at various sequence identity cutoffs. To make comparisons easier in these plots, the results of Water, which marked the highest Q3 and Q8 measured by sequence alignments in Figure 2, are shown again for reference. Structure alignment methods applied in this experiment were FAST, TM-align, and SARST. All the Q3/8 values obtained with them are higher than those obtained with sequence alignments, especially at low identities. Compared with Q3, the differences between structural and sequence alignments were much more significant in Q8. Structural alignments had a better capability of detecting the residue-residue equivalences between structural homologs than sequence alignments, even when they had distant evolutionary relationships. Thus, structural alignments should be more suitable than sequence alignments for estimating the limit of SSP.

**Figure 4 biomolecules-11-01627-f004:**
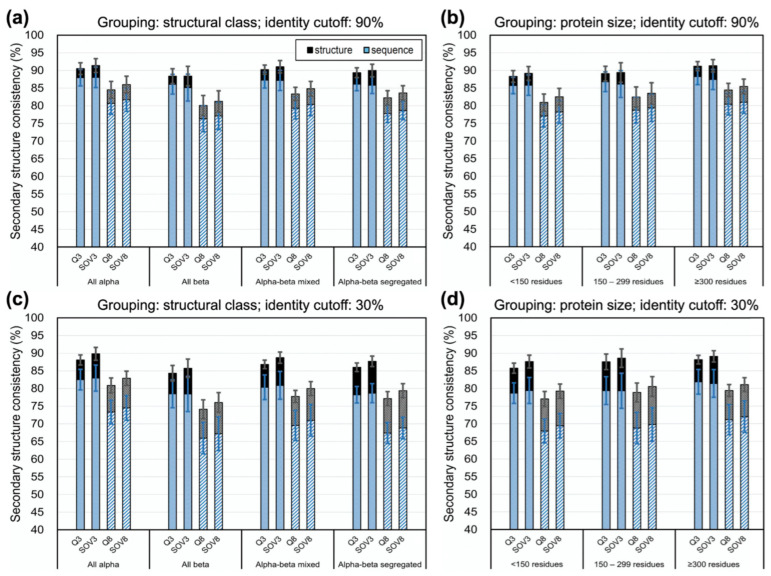
Secondary structural consistency for structural homologs of different structural classes or sizes. (**a**) The secondary structural consistency for homologs belonging to different structural classes and sharing <90% identities. (**b**) The consistency for homologs having different sizes and sharing <90% identities. (**c**) The consistency for homologs belonging to different structural classes and sharing <30% identities. (**d**) The consistency for homologs having different sizes and sharing <30% identities. In these plots, the secondary structural consistencies computed by sequence alignments (method: PSI-BLAST) and structural alignments (method: FAST) are drawn as blue and black bars, respectively; because the latter are all higher than the former, only black caps are visible. PSI-BLAST represents sequence alignment methods because most SSP algorithms work depending on it. FAST represents structure alignment methods because it reported the highest Q3/8 between SCOP homologs (Figure 3).

**Figure 5 biomolecules-11-01627-f005:**
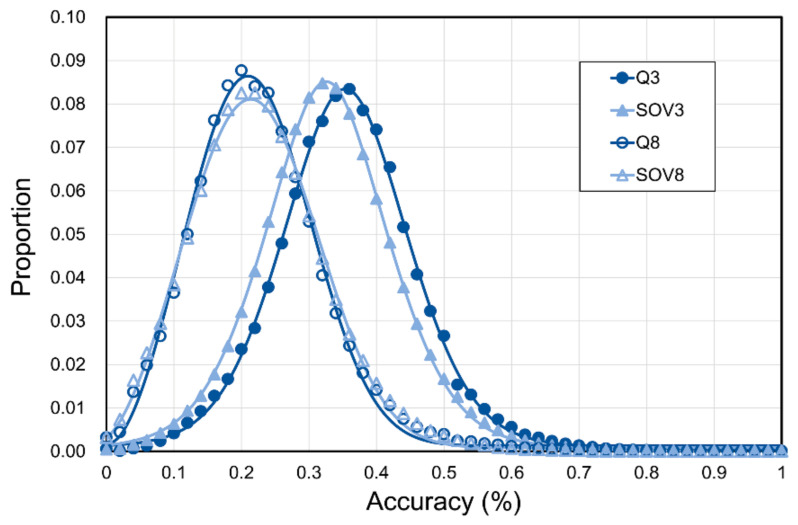
Distribution of secondary structural consistency between random protein pairs. One million rounds of random pairing were performed to compute the distribution of secondary structural consistencies between proteins to mark the lower bound of SSP accuracy. Two proteins were randomly selected from the SCOP-2.07 dataset in each round, and the secondary structural consistency between them was measured with the Q and SOV scores. The average SOV3 of these random pairs was lower than the average Q3 (SOV3 = 32% versus Q3 = 35%), different from the results of the first paper of SOV (SOV3 = 37% versus Q3 = 35%) [62], because the algorithm of SOV applied in this study was an updated version (v’99) [66], the value of which is lower than the first defined SOV (v’94; see Table I of [66] for comparisons).

**Figure 6 biomolecules-11-01627-f006:**
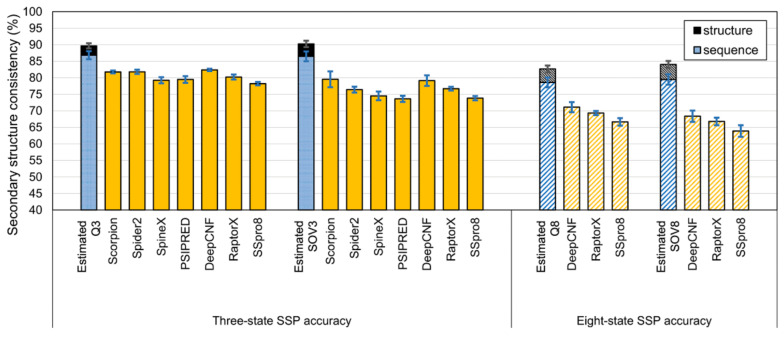
Accuracies of state-of-the-art SSP methods evaluated with datasets sharing <90% sequence identities. This test utilized the independent query sets prepared by Yang et al. for SSP evaluations [1] and the SCOP720 query set. The source of reference sequences for PSSM generation was UniRef90-2015. The blue and black bars indicate the SSP upper limits estimated at a <90% sequence identity cutoff by sequence and structural alignments, respectively. The sequence alignment algorithm used to make those estimates was PSI-BLAST, the PSSM generator of all the tested SSP methods. These results reveal that, when the homology of the SSP query and reference sequences was <90% identity, the accuracy of current SSP methods has not reached the limit estimated by sequence alignments of homologs sharing <90% sequence identities.

**Figure 7 biomolecules-11-01627-f007:**
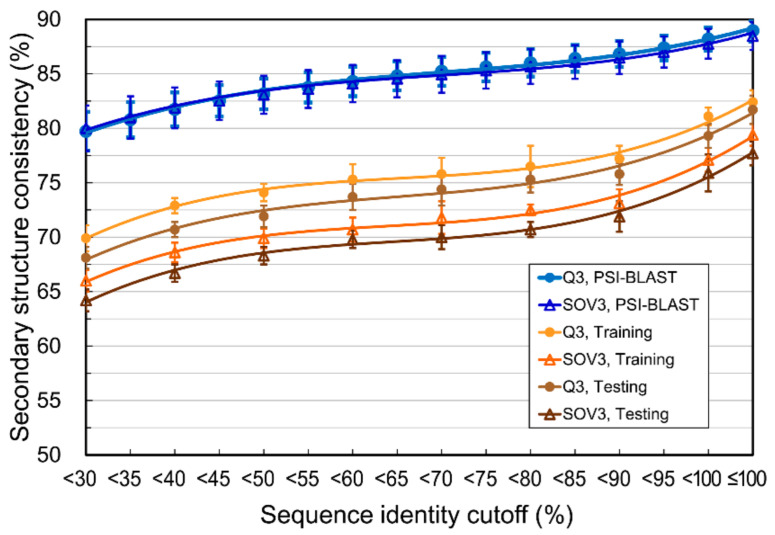
The accuracy of the current SSP methodology applied with restricted homology between developmental and operational datasets. For avoiding the side effects of sequence redundancy, which might cause information leakage and overfitting for a predictor, it is typical for computational biology studies to restrict protein datasets with some sequence homology cutoff. We speculated that the SSP limits estimated based on homologs with different homology cutoffs (blue curves, adopted from the result of PSI-BLAST in Figure 2c and Appendix A) would draw the accuracy upper bound for SSP methods trained and operated with datasets where the homology of proteins is restricted. The protein materials used here were prepared so that the sequence identities within and between the query and PSSM reference sets were lower than the given cutoffs. As expected, as the homology decreased, the accuracy of SSP models trained and operated with such proteins decreased. This experiment was repeated 10 times by random sampling.

**Figure 8 biomolecules-11-01627-f008:**
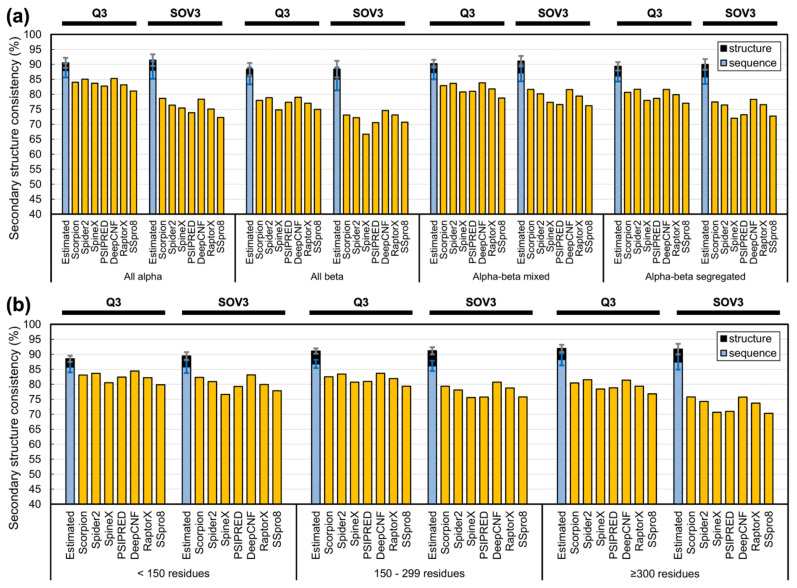
Accuracy of state-of-the-art SSP methods for proteins of different (**a**) structural classes or (**b**) sizes. The upper limit of SSP accuracy estimated by sequence and structural alignments at 90% identity cutoff of homologs are respectively indicated by blue and black bars. No method exceeded the estimated limits in any class or size group, and most methods met the greatest challenge in predicting all-beta proteins. SSP accuracy decreased a little as the protein size increased. Since the same tendencies were observed in three- and eight-state predictions, this figure only displays Q3 and SOV3. See Appendix A for Q8 and SOV8 results.

**Figure 9 biomolecules-11-01627-f009:**
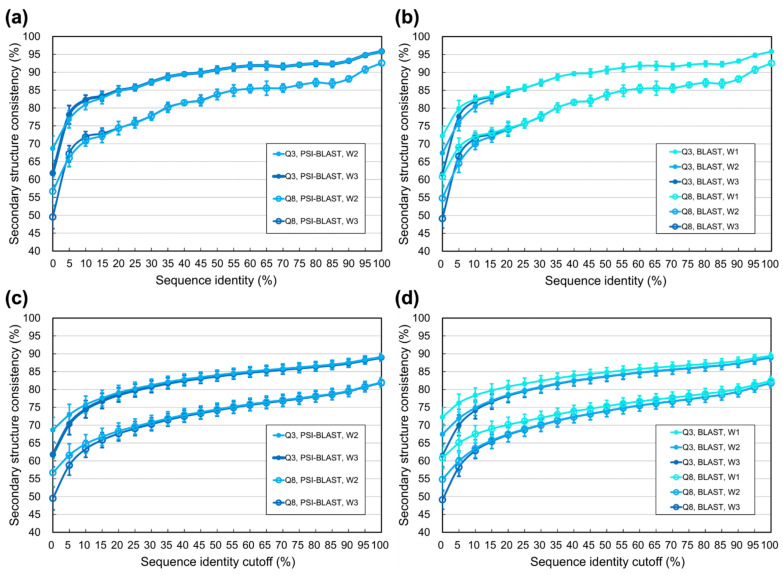
Secondary structural consistency measured by PSI-BLAST and BLAST with different word sizes. (**a**) Secondary structural consistencies between SCOP family homologs measured by PSI-BLAST at different identity levels. As illustrated by the color codes, word sizes 3 and 2 were applied. The default word size of PSI-BLAST was 3, but changing it to 2 increased the measured secondary structural consistency between homologs, especially at low identities. (**b**) Secondary structural consistencies between SCOP family homologs measured by the classic BLAST at different identity levels. The BLAST accepted word sizes 3 and 2 in the database searching mode (blastall) and word size 1 in the pairwise alignment mode (bl2seq). A small word size notably increased the measured secondary structural consistency. (**c**), (**d**) Secondary structural consistencies between SCOP family homologs measured by (PSI-)BLAST at different identity cutoffs. Results obtained according to identity cutoffs make the effects of word size more observable.

**Figure 10 biomolecules-11-01627-f010:**
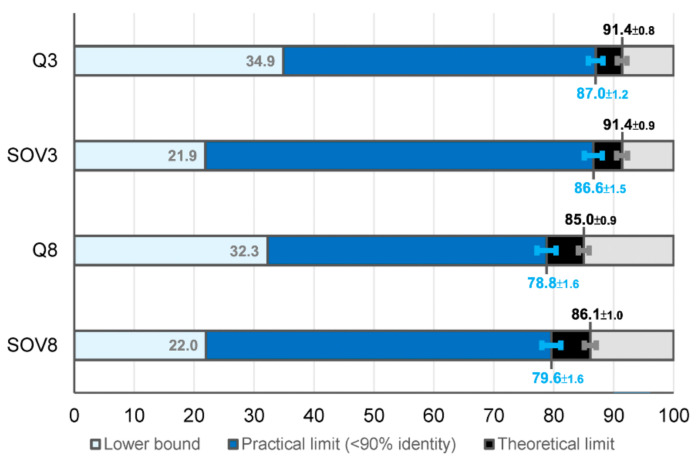
Summary of the estimated limits of SSP accuracy. Both the lower limits (light blue strips) and upper limits of three-/eight-state SSP accuracies are illustrated. The practical upper limits of accuracies (blue strips to the right) were estimated by PSI-BLAST sequence alignments with the word size 2 (see Figure 9 and Appendix A), and the theoretical limits (black strips to the right) were estimated by structural alignments (Figure 3 and Appendix A).

**Table 1 biomolecules-11-01627-t001:** Performance pretest results of several state-of-the-art SSP methods.

SSE Set	Method	Datasets: TS115 vs. UniRef90-2015		Datasets: CASP12 vs. UniRef90-2015	
Three-state	**Measure**	**Q3 (reported)**	**Q3**	**Q3 (reported)**	**Q3**
	Scorpion ^1^	0.817	0.815	0.805	0.823
	Spider2 ^1^	0.819	0.819	0.798	0.812
	SpineX ^1^	0.801	0.800	0.769	0.783
	Psipred ^1^	0.802	0.801	0.780	0.783
	DeepCNF ^1^	0.823	0.819	0.821	0.828
	RaptorX ^2^	0.812	0.807	0.791	0.793
	SSpro8^2^	0.795	0.788	0.776	0.779
Eight-state	**Measure**	**Q8 (reported)**	**Q8**	**Q8 (reported)**	**Q8**
	DeepCNF ^1^	0.720	0.703	0.730	0.728
	RaptorX ^2^	0.697	0.698	0.651	0.694
	SSpro8 ^1^	0.680	0.671	0.690	0.675

^1^ Obtained from [1], where the query datasets were TS115 and CASP12, and the PSSM reference dataset was UniRef90-2015. ^2^ Obtained from [55], where the query datasets were CullPDB and CASP11, and the PSSM reference dataset was UniRef90-2015.

**Table 2 biomolecules-11-01627-t002:** Three-state secondary structural consistency measured by sequence/structure alignments for homologs identified from PDB.

Identity (%)	Yang et al. [1]	PSI-BLAST (Avg ± Std)	FAST (Avg ± Std)	Identity Threshold (%)	Yang et al. [1]	PSI-BLAST (Avg ± Std)	FAST (Avg ± Std)
100	96.8	97.1 ± 0.3	97.1 ± 0.3	100	96.8	97.1 ± 0.3	97.1 ± 0.3
90‒100	96.2	95.6 ± 0.4	95.7 ± 0.4	≥90	96.8	96.1 ± 0.3	96.2 ± 0.4
80‒90	95.1	93.5 ± 1.3	94.0 ± 1.2	≥80	96.6	95.6 ± 0.5	95.8 ± 0.5
70‒80	95.8	92.5 ± 1.6	93.2 ± 1.3	≥70	96.5	95.4 ± 0.6	95.6 ± 0.5
60‒70	94.7	92.9 ± 0.8	93.4 ± 0.7	≥60	96.3	95.2 ± 0.6	95.5 ± 0.6
55‒60	94.2	92.6 ± 0.7	93.1 ± 0.8	≥55	96.2	95.0 ± 0.6	95.3 ± 0.6
50‒55	93.1	92.3 ± 0.6	92.8 ± 0.6	≥50	96.0	94.8 ± 0.6	95.1 ± 0.6
45‒50	92.6	91.8 ± 0.9	92.6 ± 0.8	≥45	95.7	94.6 ± 0.6	94.9 ± 0.6
40‒45	91.0	90.1 ± 1.3	91.2 ± 1.3	≥40	95.1	94.1 ± 0.7	94.5 ± 0.7
35‒40	89.9	89.6 ± 0.8	90.8 ± 0.9	≥35	94.6	93.6 ± 0.7	94.1 ± 0.7
30‒35	87.7	87.8 ± 0.8	90.2 ± 0.4	≥30	93.4	92.4 ± 0.7	93.3 ± 0.6
25‒30	84.2	84.2 ± 0.5	89.0 ± 0.4	≥25	90.7	89.1 ± 0.6	91.6 ± 0.5
20‒25	81.1	81.4 ± 0.3	88.2 ± 0.2	≥20	87.0	86.0 ± 0.5	90.2 ± 0.4
15‒20	75.9	77.5 ± 0.4	87.2 ± 0.2	≥15	84.0	83.8 ± 0.4	89.4 ± 0.3
<10‒15	71.4	72.1 ± 0.9	85.7 ± 0.3	≥10	83.7	80.8 ± 0.5	88.5 ± 0.3

**Table 3 biomolecules-11-01627-t003:** Eight-state secondary structural consistency measured by sequence/structure alignments for homologs identified from PDB.

Identity (%)	PSI-BLAST (Avg ± Std)	FAST (Avg ± Std)	Identity Threshold (%)	PSI-BLAST (Avg ± Std)	FAST (Avg ± Std)
100	94.8 ± 0.4	94.9 ± 0.4	100	94.8 ± 0.4	94.9 ± 0.4
90‒100	92.2 ± 0.7	92.6 ± 0.7	≥90	93.1 ± 0.5	93.4 ± 0.5
80‒90	88.5 ± 2.0	89.6 ± 1.7	≥80	92.3 ± 0.7	92.7 ± 0.7
70‒80	86.7 ± 2.8	88.1 ± 2.2	≥70	92.0 ± 0.9	92.4 ± 0.8
60‒70	87.2 ± 1.3	88.4 ± 1.1	≥60	91.6 ± 0.9	92.1 ± 0.9
55‒60	86.2 ± 1.8	87.3 ± 1.8	≥55	91.2 ± 1.0	91.7 ± 0.9
50‒55	85.8 ± 0.5	87.0 ± 0.5	≥50	90.8 ± 1.0	91.4 ± 0.9
45‒50	85.3 ± 1.2	86.8 ± 1.1	≥45	90.4 ± 1.0	91.1 ± 0.9
40‒45	82.6 ± 2.0	84.6 ± 1.9	≥40	89.6 ± 1.1	90.4 ± 1.0
35‒40	81.6 ± 1.2	83.9 ± 1.3	≥35	88.6 ± 1.1	89.6 ± 1.1
30‒35	78.9 ± 1.1	82.7 ± 0.5	≥30	86.7 ± 1.1	88.2 ± 0.9
25‒30	74.0 ± 0.7	80.9 ± 0.6	≥25	81.6 ± 0.9	85.3 ± 0.8
20‒25	70.7 ± 0.4	79.7 ± 0.3	≥20	77.2 ± 0.7	83.0 ± 0.6
15‒20	65.9 ± 0.4	78.3 ± 0.3	≥15	74.2 ± 0.6	81.8 ± 0.5
<10‒15	59.7 ± 1.0	76.5 ± 0.5	≥10	70.5 ± 0.7	80.4 ± 0.5

## Data Availability

All data generated or analyzed during this study are included in this published article and its Appendix A.

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
