# Peer review of "Discovering the Ultimate Limits of Protein Secondary Structure Prediction"

_biomolecules, 2021, doi:10.3390/biom11111627_

Round 1

Reviewer 1 Report

The work presented relates to an old subject: What is the theoretical limit of the prediction of secondary structures? . This research shows that this issue is still relevant today.

By an efficient approach and a lot of work these researchers demonstrate -as well for 3 states as for 8 states- that yes, it is interesting to know what is this theoretical limit, but also the practical limit.

To do this, the authors use several datasets and especially several different software to assess the relevance of their methodolgy.

I was impressed by the amount of work done and the rigor of the analysis. The paper is long to read because it is in this it makes many specific analysis. The results presented in a synthetic manner in Figure 10 clearly show the interest of the analysis presented by these Taiwanese researchers.

The work carried out being particularly rigorous and well designed and I have few questions.

(i) The first question will be related to the secondary structure assignment. It would be good to know which version of DSSP was used. In fact, around 2015, the order of assignment of the propellers was changed. This will result in few modifications for the 3-states, but a little more for the 8-states.

(ii) Likewise, the majority of prediction approaches have been trained on the old DSSP version, if I am not mistaken. Can it have an impact on the evaluation of 8-states prediction?

(iii) As a reader, I miss the precise occurrence of these 8 states and also the specific prediction of each of its states.  Indeed, it seems obvious that the prediction of the alpha helices, of the beta sheets, of the loops, perhaps of beta-turn is correct enough. On the other hand, the pi-helix and the beta bridge should be very badly predicted, even not predicted, as it was the case with the software SSpro 8 in the past. It would be good to have some indications on this.

(iv) Another question of analysis. Is there a difference in both practical and theoretical prediction between the accessible and non-accessible parts? Again, the reader expects a positive for buried region, is it the case?

(v) Are the regions with the highest (normalized) B-factors theoretically the regions that would be poorly predicted?

(vi) A discussion on disorder could be good.

(vii) As noted before, theoretical limit was searched due to dynamics of protein structures. In PMID: 32946990 DOI: 10.1016/j.biochi.2020.09.006, it is showed that near 5% of initial secondary structure assignment drastically changed (more than50% of the times, e.g. from a helical structure in a crystal to a loop structure in molecular dynamics) and that they will be better predicted with this new assignment. Can it impact these results?

(viii) The use of PSIBlast and of course expected by cons it is noted that classic blast is used but I have not seen it in figure 2 while it is noted in the legend. Did I miss something?

(ix) a General schema (pipeline) to present the methodology could be of great help

To conclude, it is therefore necessary to discuss the analysis of the results a little more according to the local structures such as helices, strand, loops and other features, e.g. their accessibility and their flexibility, and the paper will then be perfect.

Reviewer 2 Report

Ho et al provides a comprehensive re-estimation of the ultimate limit of protein secondary structure prediction. They have employed structural alignment across different structural folds and examined the consistency in secondary structure between aligned structures. Overall results are quite convincing. Here are some comments.

1) It should be mentioned that another factor limiting the secondary structure prediction is the consistency of secondary structure assignment. This work is mostly based on DSSP assignment. A different assignment method may yield a different picture [e.g. Proteins 71:61–67 (2008)].

2) Recent 8-state prediction can achieve 76-77% accuracy, rather than 76% mentioned in this work. [Bioinformatics, 35: 2403–2410 (2019)].

3) With the success of AlphaFold2, it seems that the deep learning can now achieve highly accurate prediction of both secondary and tertiary structure. It should be mentioned here as well.

Reviewer 3 Report

In this manuscript, Ho et al. revise the existing approaches to predicting protein secondary structure, and in particular, the related estimates of their accuracy limits. Through a series of computational experiments, they refine the existing upper bounds and compute lower bounds on those limits for the cases of three-and eight-stage predictions applying two distinct accuracy measures (all together resulting in the Q3, Q8, SOV3 and SOV8 accuracy measures). In addition, they propose to distinguish between theoretical and practical upper limits established using structural and sequence alignments, respectively. Finally, they suggest a few potential ways for reducing the currently observed gaps between those limits.

The research framework looks consistent; the obtained results are appropriately illustrated in figures and tables. The idea to consider theoretical upper limits computed based on structural alignments seems relevant and might help to further improve the methods for predicting secondary structure of proteins. In my opinion, this paper is suitable for publication in Biomolecules.

Round 2

Reviewer 1 Report

The authors answered all my questions perfectly and in a particularly thorough manner. An excellent paper.